# JORGE: APPROXIMATE PRECONDITIONING FOR GPU-EFFICIENT SECOND-ORDER OPTIMIZATION

## ABSTRACT

Despite their better convergence properties compared to first-order optimizers, second-order optimizers for deep learning have been less popular due to their significant computational costs. The primary efficiency bottleneck in such optimizers is matrix inverse calculations in the preconditioning step, which are expensive to compute on GPUs. In this paper, we introduce Jorge, a second-order optimizer that promises the best of both worlds – rapid convergence benefits of second-order methods, and high computational efficiency typical of first-order methods. We address the primary computational bottleneck of computing matrix inverses by completely eliminating them using an approximation of the preconditioner computation. This makes Jorge extremely efficient on GPUs in terms of wall-clock computation. Further, we describe an approach to determine Jorge's hyperparameters directly from a well-tuned SGD baseline, thereby significantly minimizing tuning efforts. Our empirical evaluations demonstrate the distinct advantages of using Jorge, outperforming state-of-the-art optimizers such as SGD, AdamW, and Shampoo across multiple deep learning models, both in terms of sample efficiency and wall-clock time.

## 1 INTRODUCTION

Stochastic optimization methods such as stochastic gradient descent (SGD) (Robbins & Monro, 1951) and *Adam* (Kingma & Ba, 2015) are the de-facto standard for optimizing the objective function in the training of deep neural networks. These first-order optimization methods are relatively inexpensive in terms of their compute and memory requirements, and hence extremely popular. Second-order optimization methods typically have better convergence properties (fewer epochs to reach target validation metrics) than those of first-order methods. However, they are considerably slower in terms of per-iteration (per-batch) wall-clock times for training than first-order methods. This is because they often use a preconditioner, which multiplies the gradient by a matrix before taking a step. Computing these preconditioners requires performing matrix inversions, which are highly inefficient on GPU platforms due to the iterative nature of matrix inverse algorithms and their irregular memory access patterns.

If one could develop a second-order optimizer that has better convergence than first-order methods and is on par with them in terms of wall-clock time per iteration, we could achieve the best of both worlds. In this paper, we present Jorge, a new second-order optimizer that uses an approximation for preconditioning by avoiding the calculation of the inverse of matrices in all steps. It has similar convergence properties to other second-order optimization methods but its wall-clock time per iteration is similar to that of inexpensive first-order methods. This is a win-win situation, which leads to much faster total training times for several different deep learning models when compared to other state-of-the-art optimizers.

A new optimization method is most useful and promising if users do not have to spend significant time in tuning its hyperparameters. We demonstrate the process of deriving reasonable hyperparameters for Jorge from a well-tuned SGD baseline with minimal effort. Interestingly, these derived hyperparameters match the generalization of SGD and even improve it in many cases! Note that we use SGD over other adaptive optimizers such as Adam because prior research has shown that SGD often outperforms adaptive methods in terms of generalization (Wilson et al., 2017). In our experiments across different network architectures, we demonstrate that Jorge performs better than

two widely adopted first-order optimizers, SGD and AdamW, both in terms of sample efficiency and overall wall-clock times for convergence. Additionally, we demonstrate comparable sample efficiency to Shampoo (Gupta et al., 2018), a state-of-the-art second-order optimizer, while achieving faster convergence times.

This paper makes the following important contributions:

- A new second-order optimizer that avoids matrix inverse calculations when computing the preconditioner, making it extremely efficient on GPUs. This results in per-iteration wall-clock times within 5-10% of those of first-order optimizers such as SGD and AdamW, while matching the sample efficiency of Shampoo, a second-order optimizer. For training ResNet-50 on ImageNet, we demonstrate improvements of nearly 25% in the total training wall-clock time over SGD.
- We show that reasonable hyperparameter configurations for Jorge can be easily bootstrapped from those of a well-tuned SGD baseline without extensive hyperparameter tuning that would require full training runs. These settings result in either similar and in many cases, even better generalization than that of SGD!
- Most second-order optimizers need to exploit complex parallelism requiring multiple GPUs to get their total training times to be faster than those of first-order optimizers. Since Jorge is highly efficient, it can be run locally on each GPU and still outperform highly optimized parallel implementations of second-order optimizers.

## 1.1 RELATED WORK

There have been several research efforts to develop computationally tractable second-order optimizers for deep learning. Martens (2010) proposes Hessian-free optimization, which exploits conjugate gradient (CG) to directly compute Hessian-vector products without explicitly computing the Hessian. Since CG requires multiple iterations, there has been subsequent work on reducing this cost (Erdogdu & Montanari, 2015). Several optimizers based on the L-BFGS method have also been proposed that approximate Hessian-vector products from the history of past gradients, again without explicitly computing the Hessian (Berahas et al., 2016; Bollapragada et al., 2018; Wang et al., 2017).

Most state-of-the-art second-order optimizers rely on block-diagonal approximations of the Hessian to reduce the computational and memory requirements. The "blocks" typically correspond to substructures in the neural network, like a layer or a parameter tensor. Some recent methods in this category include Shampoo (Gupta et al., 2018), K-FAC (Martens & Grosse, 2015; Grosse & Martens, 2016), K-BFGS (Goldfarb et al., 2020) and the GGT method (Agarwal et al., 2019). However, these methods need to compute the inverse of their approximate Hessian matrices, which can be expensive to compute even with the block-diagonal approximations. As we show later in Section 5, Jorge outperforms one such optimizer, Shampoo, by nearly 37% in terms of the total wall-clock time for training ResNet50 on ImageNet. Closely related to Jorge is a line of work that exploits the Sherman-Morrison based Matrix identity to approximate the update steps in K-FAC without computing any matrix inverses (Mozaffari et al., 2023; Zhang et al., 2023; Tang et al., 2021).

To mitigate the large computational costs of matrix inverses, researchers have also proposed parallel implementations of second-order optimizers, which aim to distribute the work of the optimizer across multiple GPUs. Several efforts focus on developing efficient parallel implementations of the K-FAC optimizer (Pauloski et al., 2020; 2021; Osawa et al., 2019; 2020; Ueno et al., 2020; Shi et al., 2021). On the other hand, Shi et al. (2023) and Anil et al. (2021) aim to accelerate the Shampoo (Gupta et al., 2018) optimizer via parallelism. Anil et al. (2021) present a heterogeneous solution that offloads the computation of the inverses to the CPU. Even though we implement Jorge without any multi-GPU parallelism, we demonstrate that its performance is better than one of the state-of-the art parallel optimizers – Distributed Shampoo (Shi et al., 2023).

## 2 BACKGROUND

Second-order optimizers make use of both the gradients and curvature (second derivatives) of the loss function. By considering the curvature, second-order methods can approximate the loss function

more accurately than first-order optimizers, and thus reduce the number of iterations required for convergence. Most second-order optimizers approximate the Newton step shown in Equation 1.

$$\theta_t = \theta_{t-1} - H_t^{-1} \, G_t \tag{1}$$

Hessian at timestep t     gradients at timestep t

This equation can be derived by minimizing a second-order Taylor's approximation of the loss function at $\theta_t$. This step of multiplying the gradients with $H_t^{-1}$ is called preconditioning, and $H_t^{-1}$ is often referred to as a preconditioner.

Instead of using the actual Hessian, optimizers typically use positive semi-definite approximations of the Hessian (Schraudolph, 2002; Amari, 1998) to account for the non-convexity of the training objective (Vinyals & Povey, 2012; Botev et al., 2017; Roux et al., 2007; Martens & Grosse, 2015; Desjardins et al., 2015). Our proposed optimizer, Jorge, belongs to a class of methods called "adaptive optimizers", which use the inverse of the gradient covariance matrix (or the empirical Fisher matrix) to precondition gradients. Examples of adaptive second-order optimizers include the full matrix version of Adagrad (Duchi et al., 2011) and Shampoo (Gupta et al., 2018). Note that several first-order adaptive optimizers have also been proposed in literature, which only use the diagonal elements of the covariance matrix. Popular examples include Adam (Kingma & Ba, 2015) and RMSProp. Jastrzebski et al. (2018); Sagun et al. (2018); Zhu et al. (2019) provide justification for the usage of the gradient covariance matrix as an approximation of the Hessian.

## 3 APPROXIMATE PRECONDITIONING IN JORGE

As described in Section 1.1, the primary efficiency bottleneck in state-of-the-art second-order optimizers such as K-FAC (Martens & Grosse, 2015) and Shampoo (Gupta et al., 2018) is the matrix inverse computations performed to calculate the preconditioners. To overcome this limitation, we introduce Jorge, an efficient, adaptive, second-order optimizer tailored for GPU execution. Jorge's formulation eliminates computing explicit matrix inversions, and is solely comprised of matrix multiplications and additions, which are highly optimized on GPUs. This results in Jorge's wall-clock time per iteration to be on par with those of first-order optimizers, while also having faster convergence properties typical of a second-order optimizer.

We propose Jorge as an enhancement of Shampoo (Gupta et al., 2018), another adaptive second-order optimizer. We first describe Shampoo's optimizer algorithm at a high level before describing Jorge's optimizer algorithm. Note that, throughput this section, we discuss Shampoo and by extension Jorge, within the context of a single layer. Application to multiple layers simply involves repeating the same steps for their parameters.

Following Gupta et al. (2018), let us assume that the parameters, $\theta$, of a single layer are organized in a two-dimensional (2D) $m \times n$ matrix (N-dimensional parameter tensors, like those found in convolution layers are typically collapsed into 2D matrices, in practice). Shampoo maintains the second-order curvature information of the loss in two matrices – $L_t$ (size $m \times m$) and $R_t$ (size $n \times n$), which are called the left and right preconditioners, respectively. It iteratively updates the preconditioners from the current gradient information as shown in the equation below (for the left preconditioner):

left preconditioner at timestep t and t-1

$$L_t = \beta_2 \, L_{t-1} + (1 - \beta_2) \, G_t \, G_t^T \tag{2}$$

smoothing parameter     gradients at timestep t

Algorithm 1 shows how the preconditioners are used in Shampoo. Additional terms used in the algorithm are defined as follows. $\beta_1$ and $\beta_2$ are smoothing parameters for the exponential moving average (EMA) of the momentum and preconditioners. $\tilde{G}_t$ is the preconditioned gradients at timestep $t$. $m_t$ is the EMA of the preconditioned gradients, and $\eta_t$ is the learning rate at timestep

$t$. Lines 5–8 of Algorithm 1 show how the Shampoo optimizer iteratively updates the left and right preconditioners from the current gradients' information. Line 11 illustrates the preconditioning step, wherein the gradients is multiplied by $L_t^{\frac{-1}{4}}$ and $R_t^{\frac{-1}{4}}$ on the left and right, respectively. The preconditioning step produces the preconditioned gradients, $\tilde{G}_t$, which minimize the loss faster than the raw gradients. Finally, we update the momentum estimate of the preconditioned gradients (line 14), and then use the momentum to update the weights (line 15). The matrix inverse computation in the preconditioning step (line 11) is the primary efficiency bottleneck in Shampoo, and is exactly what we want to optimize in Jorge.

| **Algorithm 1** Shampoo | **Algorithm 2** Jorge compared to Shampoo |
|---|---|
| 1: **Initialize** $\theta_0$, $L_0 = \epsilon I_m$ | 1: **Initialize** $\theta_0$, $\hat{L}_0 = \epsilon^{-\frac{1}{4}} I_m$ , $\hat{R}_0 = \epsilon^{-\frac{1}{4}} I_n$ |
| 2: $\qquad\qquad\quad R_0 = \epsilon I_n$ | 2: |
| 3: **for** t=1 ,..., T **do** | 3: **for** t=1 ,..., T **do** |
| 4: $\quad$ **Update Preconditioners:** | 4: $\quad$ **Update Preconditioners:** |
| 5: $\quad L_t = \beta_2 L_{t-1}$ | 5: $\quad X_L = \hat{L}_{t-1}^4 G_t G_t^T$ |
| 6: $\qquad\quad +(1-\beta_2)G_t G_t^T$ | 6: $\quad \hat{L}_t = \beta_2^{\frac{-1}{4}} \hat{L}_{t-1} \left( I_m - \dfrac{(1-\beta_2)}{4\beta_2} X_L + \dfrac{5(1-\beta_2)^2}{32\beta_2^2} X_L^2 \right)$ |
| 7: $\quad R_t = \beta_2 R_{t-1}$ | |
| 8: $\qquad\quad +(1-\beta_2)G_t^T G_t$ | 7: $\quad X_R = \hat{R}_{t-1}^4 G_t^T G_t$ |
| 9: | 8: $\quad \hat{R}_t = (\beta_2')^{\frac{-1}{4}} \hat{R}_{t-1} \left( I_n - \dfrac{(1-\beta_2')}{4\beta_2'} X_R + \dfrac{5(1-\beta_2')^2}{32(\beta_2')^2} X_R^2 \right)$ |
| | 9: |
| 10: $\quad$ **Precondition Gradients:** | 10: $\quad$ **Precondition Gradients:** |
| 11: $\quad \tilde{G}_t = L_t^{\frac{-1}{4}} G_t R_t^{\frac{-1}{4}}$ | 11: $\quad \tilde{G}_t = \hat{L}_t G_t \hat{R}_t$ |
| 12: | 12: |
| 13: $\quad$ **Update Weights:** | 13: $\quad$ **Update Weights:** |
| 14: $\quad m_t = \beta_1 m_{t-1} + (1-\beta_1)\tilde{G}_t$ | 14: $\quad m_t = \beta_1 m_{t-1} + (1-\beta_1)\tilde{G}_t$ |
| 15: $\quad \theta_t = \theta_{t-1} - \eta_t m_t$ | 15: $\quad \theta_t = \theta_{t-1} - \eta_t m_t$ |
| 16: **end for** | 16: **end for** |

In Algorithm 2, we show the functioning of Jorge side-by-side with Shampoo for the same 2D $m \times n$ parameter matrix of a single layer. The core idea behind Jorge is to approximate the computation of $L_t^{\frac{-1}{4}}$ and $R_t^{\frac{-1}{4}}$ in Shampoo (line 11 of Algorithm 1) in a GPU-efficient manner. In order to do this, we modify the computation in both lines 5–8 and line 11 of Algorithm 1. Just like Shampoo, Jorge also maintains two preconditioners, which we refer to as $\hat{L}_t$ and $\hat{R}_t$ in Algorithm 2. However, Jorge's preconditioners are an approximation of the inverse fourth root of Shampoo's preconditioners at every iteration, i.e. $\hat{L}_t \approx L_t^{\frac{-1}{4}}$ and $\hat{R}_t \approx R_t^{\frac{-1}{4}}$. We show the remaining steps for the left preconditioner approximation, and the right preconditioner approximation can be derived similarly.

Since $\hat{L}_t \approx L_t^{\frac{-1}{4}}$, we can say that $L_t \approx \hat{L}_t^{-4}$, and $L_{t-1} \approx \hat{L}_{t-1}^{-4}$. We substitute $L_t$ and $L_{t-1}$ on both sides of Equation 2, which gives us:

$$\hat{L}_t^{-4} = \beta_2 \hat{L}_{t-1}^{-4} + (1-\beta_2)G_t G_t^T \tag{3}$$

$$\implies \hat{L}_t = \left( \beta_2 \hat{L}_{t-1}^{-4} + (1-\beta_2)G_t G_t^T \right)^{\frac{-1}{4}}$$

$$= \beta_2^{\frac{-1}{4}} \hat{L}_{t-1} \left( I_m + \dfrac{(1-\beta_2)}{\beta_2} \hat{L}_{t-1}^4 G_t G_t^T \right)^{\frac{-1}{4}}$$

$$= \beta_2^{\frac{-1}{4}} \hat{L}_{t-1} \left( I_m + \dfrac{(1-\beta_2)}{\beta_2} X_L \right)^{\frac{-1}{4}} \tag{4}$$

$$\underbrace{\hat{L}_{t-1}^4 G_t G_t^T}_{\text{(line 5, Algorithm 2)}}$$

Next, we get rid of the inverse computation in Equation 4 by employing the binomial series expansion on the expression in parenthesis. The binomial theorem for negative exponents suggests that for a square matrix $A \in \mathbb{R}^{m \times m}$, provided $\|A\| < 1$ and $p > 0$, where $\|.\|$ is a valid matrix norm, the following is true:

$$(I_m + A)^{-p} = \sum_{r=0}^{\infty} (-1)^r \frac{p(p+1)(p+2)...(p+r-1)}{r!} A^r \tag{5}$$

Subsituting $A = \frac{(1-\beta_2)}{\beta_2} X_L$, and $p = \frac{1}{4}$ in Equation 5 yields:

$$\left( I_m + \frac{(1-\beta_2)}{\beta_2} X_L \right)^{\frac{-1}{4}} = I_m - \frac{1}{4} \frac{(1-\beta_2)}{\beta_2} X_L + \frac{5}{32} \frac{(1-\beta_2)^2}{\beta_2^2} X_L^2 + ... \tag{6}$$

Now, replacing the expression in parenthesis in Equation 4 with its binomial series expansion in Equation 6, we remove the inverse calculation entirely as shown below:

$$\hat{L}_t = \beta_2^{\frac{-1}{4}} \hat{L}_{t-1} \left( I_m - \frac{1}{4} \frac{(1-\beta_2)}{\beta_2} X_L + \frac{5}{32} \frac{(1-\beta_2)^2}{\beta_2^2} X_L^2 + ... \right) \tag{7}$$

Note that the binomial expansion is an infinite series and thus intractable. In practice, we have found that ignoring the cubic and higher powers of this expansion does not degrade the sample efficiency of Jorge in comparison to Shampoo (See Section 5). Hence we drop the higher-order terms in Equation 7, which gives us line 6 of Algorithm 2. Notice how our preconditioner update step is composed entirely of matrix-matrix multiplications and additions, which are highly efficient to compute on GPUs, thereby making Jorge more compute-efficient than other second-order optimizers. After updating the preconditioners, we precondition the gradients by multiplying them with $\hat{L}_t$ and $\hat{R}_t$ on the left and right (line 11). Unlike Shampoo, we do not have to invert our preconditioners because, by definition, they are an approximation of the inverse fourth roots of Shampoo's preconditioners. Finally, the weight update step in lines 14 and 15 is identical to Shampoo.

Note that Equation 5 is only valid for $\|A\| < 1$, and therefore for $\|\frac{(1-\beta_2)}{\beta_2} X_L\| < 1$. To ensure this, Jorge dynamically adjusts $\beta_2$ (and $\beta_2'$ for the right preconditioner) in each iteration such that the above constraint is met. We discuss this in detail in Appendix A.1.

To improve performance, most second-order optimizers, including K-FAC and Shampoo, typically compute their preconditioners at regular intervals, instead of every iteration. Following suit, we also allow infrequent preconditioner updates for Jorge, with the interval kept as a user-configurable hyperparameter. In the iterations where we do not update the preconditioners, we simply reuse the preconditioners from the previous iteration.

As empirical evidence of the efficacy of our approximation we measured the per-iteration times of SGD, Jorge and AdamW for training ResNet-50 (He et al., 2016b) and DeepLabv3 (Chen et al., 2017), and found Jorge to be 21–26% faster than Shampoo, and within 10% of SGD (more details in Appendix A.2).

## 4 BOOTSTRAPPING JORGE'S HYPERPARAMETERS FROM SGD

A new optimizer such as Jorge would be useful in practice only if it does not require rigorous hyperparameter tuning to achieve a desired level of generalization on a given training task. Arguably, an important reason behind the popularity of SGD is the existence of various heuristics for deciding hyperparameters configurations quickly that can achieve decent generalization. In this section, we demonstrate Jorge's ability to be an effective drop-in for SGD. We propose rules to deterministically bootstrap Jorge's hyperparameters from those of a well-tuned SGD baseline. We call this process "single-shot tuning". There are two implications of being able to single-shot tune Jorge's hyperparameters from a well-tuned SGD. First, it eliminates the need to explore the expensive, combinatorial search space of Jorge's hyperparameters. Second, the heuristics used to tune SGD's hyperparameters can also be transferred to Jorge.

Note that we focus on SGD over other adaptive optimizers such as Adam because prior research has demonstrated that SGD often outperforms adaptive methods in terms of generalization (Wilson et al., 2017; Zhuang et al., 2020; Keskar & Socher, 2017; Luo et al., 2019). Below, we propose some rules for transferring SGD's hyperparameters to Jorge.

**Learning Rate:** Agarwal et al. (2020) propose grafting, a technique for bootstrapping the learning rate and schedule of a new optimizer from another well-tuned optimizer. Grafting calculates the magnitude of the weight update by running a step of the well-tuned optimizer, and the direction of the weight update by running a step of the new optimizer. Using this approach, we employ grafting to directly use the learning rate of a well-tuned SGD baseline in Jorge. Integrating grafting in Jorge involves a small tweak to the weight update step in Algorithm 2 (lines 13-15), which we show in Appendix A.3. However, note that unlike Agarwal et al. (2020), we exploit grafting to adopt only the learning rate from SGD, but not the learning rate schedule (more details below).

**Weight Decay Penalty:** For regularization, in Jorge, we implement the decoupled weight decay scheme proposed by Loshchilov & Hutter (2017a), as it has been shown to generalize better than L2 regularization for adaptive optimizers. We now explain how the weight decay penalty for Jorge, $\lambda_{\text{Jorge}}$, can be bootstrapped from SGD. Let $\beta_{\text{SGD}}$ and $\lambda_{\text{SGD}}$ be the momentum factor and the weight decay penalty, respectively, of a well-tuned SGD optimizer. We propose deterministically setting $\lambda_{\text{Jorge}}$ as follows:

$$\lambda_{\text{Jorge}} = \frac{1}{1 - \beta_{\text{SGD}}} \lambda_{\text{SGD}} \tag{8}$$

Using the almost universal value of 0.9 for $\beta_{\text{SGD}}$, we set Jorge's weight decay to $10\times$ that of SGD for our experiments. While surprisingly simple, we have found this heuristic to work well across several benchmarks. In Appendix A.4, we describe the intuition behind Equation 8 in more detail.

**Learning Rate Schedule** As per Agarwal et al. (2020), grafting should allow us to borrow not only the learning rate, but also the learning rate schedule of a well-tuned SGD baseline. However, we find that certain learning rate schedules are not suitable for Jorge. In Figure 1, we plot the progression of validation metrics for training ResNet-18 (He et al., 2016a) on CIFAR-10 (Krizhevsky et al.) (left plot) and DeepLabv3 (Chen et al., 2017) on MS COCO (Lin et al., 2015) (right plot). Note that using the default learning rate schedules of SGD, which are the cosine (Loshchilov & Hutter, 2017b) and polynomial rate schedules, respectively, leads to barely any improvements in sample efficiency over SGD. Interestingly, simply switching to the step decay schedule with 2 decay steps (reducing the learning rate by $10\times$ at each step) at one-third and two-thirds of the total training epochs (total epochs same as that of the tuned SGD baseline) resolves this issue. We observe sample efficiency gains of nearly 1.4–1.8× over SGD. Therefore, across all training tasks, we opt for the step decay learning rate schedule with the aforementioned configuration. Interestingly, in certain scenarios using the default learning rate schedule of a given well-tuned SGD baseline also leads to overfitting with Jorge. We discuss this in Appendix A.5.

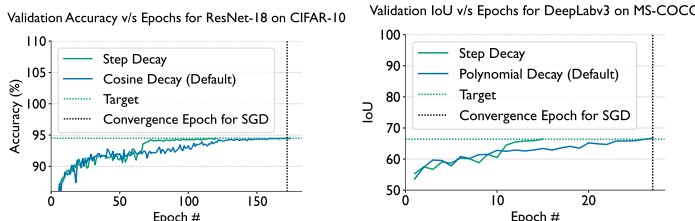

Figure 1: Comparing various learning rate schedules for Jorge. The left and right plots demonstrate the progression of validation accuracy for ResNet-18 on CIFAR-10, and validation IoU for DeepLabv3 on MS-COCO respectively.

**Preconditioner Update Frequency:** As mentioned in Section 3, Jorge has a user-configurable hyperparameter to control the frequency at which the preconditioners are updated. We suggest using a value for this hyperparameter that brings the iteration wall-clock times within 10% of SGD.

## 5 Experimental Results

In this section, we discuss the empirical experiments conducted to evaluate the efficacy of Jorge against other state-of-the-art optimizers used in deep learning.

### 5.1 Setup: Benchmarks and Metrics

Table 1 lists the training benchmarks used in our experiments, all of which are sourced from the torchvision repository (maintainers & contributors, 2016). For each benchmark, we consider two types of training runs – one where we let a given optimizer train for the maximum number of epochs specified in the repository, and the other where we only train upto the validation metrics specified in Table 1. The former helps us measure the generalization of each optimizer, whereas the latter helps us measure the sample efficiencies and total wall-clock times for training. Mask-RCNN (He et al., 2017) and DeepLabv3 (Chen et al., 2017) use ResNet-50 as their backbone. We use SGD as our baseline and also compare with AdamW, Shampoo, and a recently proposed parallel implementation of Shampoo (Shi et al., 2023),

Table 1: List of benchmarks used to evaluate Jorge against other optimizers. The validation targets for the first two tasks are the same as those used in MLPerf. For the image segmentation task, it is the same as specified in the torchvision repository.

| Training Task | Neural Network | Dataset | Batch Size(s) | Target Validation Metric |
|---|---|---|---|---|
| Image Classification | ResNet-50 | ImageNet | 256/1024 | 75.9% Accuracy |
| Object Detection | Mask-RCNN | MS-COCO 2017 | 32 | 37.7 Bbox mAP |
| Image Segmentation | DeepLabv3 | MS-COCO 2017 | 64 | 66.4 IoU |

**Choice of Hyperparameters:** For direct comparisons with SGD and AdamW, we use the default small batch sizes specified by torchvision, which are 256, 32 and 64 respectively for ResNet-50, Mask-RCNN, and DeepLabv3. To the best of our knowledge, most evaluations of second-order optimizers have been conducted at batch sizes much larger than these values. Thus, to facilitate a direct comparison with Shampoo, we also ran the ResNet-50 benchmark with a larger batch size of 1024. By doing this, we could directly borrow the hyperparameters from Shi et al. (2023), who evaluated Shampoo in a similar setting.

All the benchmarks from torchvision used in our experiments employ an SGD optimizer, pre-optimized with a well-calibrated set of hyperparameters. Accordingly, for our evaluations with SGD, we adhere to these pre-set values. For our proposed optimizer, Jorge, we adopt the single-shot hyperparameter configuration outlined in Section 4, which is derived directly from SGD's parameters. We borrow AdamW hyperparameters for the imagenet benchmarks from Heo et al. (2021). The complete list of all hyperparameters used in this study can be found in Appendix A.6.

**Evaluation Metrics:** In our evaluation of each benchmark, we record validation accuracy/IoU/mAP with respect to both number of epochs and wall-clock time. While the epoch-based measurements provide insights into the sample efficiencies of different optimizers, wall-clock time offers an understanding of their computational speed and efficiency on GPU platforms. Together, these metrics offer a comprehensive assessment of each optimizer's practical efficacy.

### 5.2 Comparative Evaluation

Rapid convergence toward a target validation accuracy is not the only goal of an optimizer. The balance between quick initial convergence and eventual generalization can dictate an optimizer's selection. For example, SGD remains the optimizer of choice in computer vision due to its better final validation accuracy, even though Adam converges faster initially. We evaluate Jorge's peak validation accuracy against SGD and AdamW across benchmarks, and detail the results in Table 2. In these experiments, we let each optimizer train for the maximum number of epochs specified in the repository. Notably, for ResNet-50 benchmarks, Jorge exceeds SGD's best validation accuracy – 76.02% vs 76.70% (large batch size), and 75.97% – 76.85% (small batch size). For the Mask-RCNN benchmark, Jorge's IoU of 38.92% represents a notable improvement over SGD's 38.3%. It's

worth highlighting that these results were achieved using the single-shot tuning strategy described in Section 4. Though DeepLabv3's performance with Jorge is marginally worse than that with SGD, the difference is within SGD's standard deviation, suggesting that small hyperparameter tweaks could bridge the gap. Notably, AdamW falls short of SGD's generalization in three out of four benchmarks but Jorge does better than SGD in three out of four benchmarks. This inconsistency in AdamW's generalization capabilities due to overfitting has piqued considerable interest and has been a focal point in several prior studies (Wilson et al., 2017; Zhuang et al., 2020; Keskar & Socher, 2017; Luo et al., 2019).

Table 2: Maximum validation accuracy ($\mu_{\pm\sigma}$) for SGD, AdamW, and Jorge across benchmarks.

| Neural Network | Batch Size | # Trials | # Epochs | SGD | AdamW | Jorge |
|---|---|---|---|---|---|---|
| ResNet-50 | 1024 | 3 | 90 | $76.02_{\pm 0.05}$ | $71.85_{\pm 0.11}$ | $\mathbf{76.70}_{\pm 0.07}$ |
| ResNet-50 | 256 | 3 | 90 | $75.97_{\pm 0.11}$ | $76.56_{\pm 0.09}$ | $\mathbf{76.85}_{\pm 0.12}$ |
| DeepLabv3 | 64 | 5 | 30 | $\mathbf{67.19}_{\pm 0.16}$ | $66.26_{\pm 0.20}$ | $67.12_{\pm 0.12}$ |
| Mask-RCNN | 32 | 5 | 26 | $38.30_{\pm 0.13}$ | $36.58_{\pm 0.11}$ | $\mathbf{38.92}_{\pm 0.10}$ |

Next, we compare the sample efficiency of Jorge to other optimizers. In this case, we only train up to the target validation metrics specified in Table 1. Figure 2 (left) showcases the progression of validation accuracy over training epochs for ResNet-50 on ImageNet with the larger batch size of 1024. For other benchmarks, we depict this progression in Figure 3. It is evident that in the context of sample efficiency, Jorge outperforms the first-order optimizers we compare with – SGD and AdamW. Across both the small (256) and large (1024) batch size training scenarios for ResNet-50, Jorge outperforms SGD by requiring around 27% fewer iterations to reach the target validation accuracy of 75.9%. The improvements in sample efficiency over SGD across other benchmarks are markedly higher – 40% for DeepLabv3, and 41% for Mask-RCNN. Again, we achieve these results by simply bootstrapping Jorge's hyperparameters from SGD, only making the changes outlined in Section 4. The improvements in sample efficiency over AdamW are similar to those over SGD. Also, AdamW falls short of achieving the target validation metric in two out of four experiments.

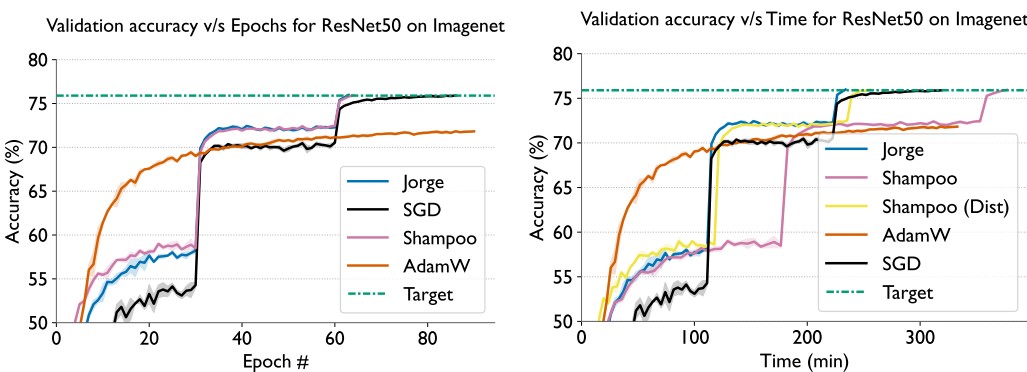

Figure 2: Validation accuracy $[\mu \pm \sigma]$ v/s epochs (left) and time (right) for the large batch size training (1024) of ResNet-50 on the ImageNet dataset (experiments run on 16 A100 GPUs).

As discussed in Section 3, we have designed Jorge to approximate Shampoo with a focus on GPU efficiency. Figure 2 (left) demonstrates that Jorge achieves the target validation accuracy in almost the same number of epochs as Shampoo (62 vs. 63). This observation strongly validates our approach and confirms that Jorge's approximations do not degrade its statistical efficiency.

Let us now turn our attention to an equally crucial metric: wall-clock time required for training. Figure 2 (right) demonstrates the progression of validation accuracy over time for the large batch size training of ResNet-50. We observe that Jorge achieves the target validation accuracy in 25% less time compared to SGD, which is a significant improvement. If we consider the serial implementation of Shampoo (pink line), it takes more total time to converge than SGD despite requiring 27% fewer epochs. This observation demonstrates the prowess of Jorge as a GPU-efficient adaptation of Shampoo: it's significantly faster than Shampoo's wall-clock time for convergence (239

minutes vs. 325 minutes), despite requiring a similar number of epochs. As noted in Section 1.1, the prevailing approach for mitigating the large overhead of preconditioning has been to develop distributed implementations of these optimizers. Within this context, Figure 2 (right) also presents the wall-clock time of a state-of-the-art parallel implementation of Shampoo (yellow line) (Shi et al., 2023). Notably, even though Jorge executes locally on each GPU, it still manages to yield a 5% speedup over the parallel version of Shampoo.

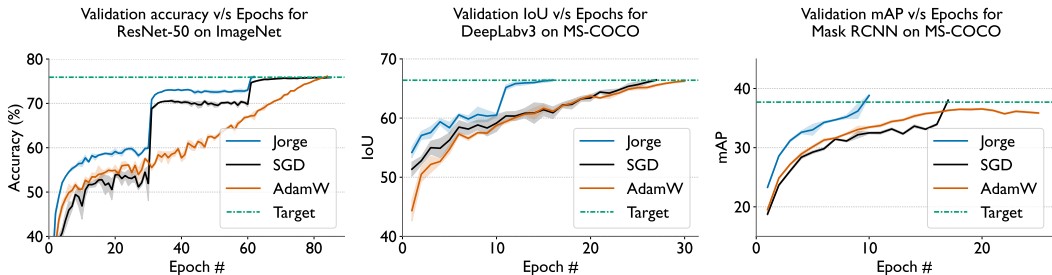

Figure 3: Validation accuracy, IoU, and mAP $[\mu \pm \sigma]$ v/s epochs for ResNet-50 on ImageNet (left) (batch size of 256), DeepLabv3 on MS-COCO (center), and Mask-RCNN on MS-COCO (right).

While a 5% improvement might seem modest, its implications are more far-reaching. Often times, AI practitioners do not have access to large numbers of GPU resources. In such resource-constrained settings, Jorge might be an ideal optimizer when parallelizing across GPUs is not an option. This also applies to environments with limited interconnect bandwidth.

Finally, we focus on the small batch size benchmarks to evaluate how Jorge's training wall-clock times compare with other first-order optimizers. We present these results in Table 3. Once again, Jorge makes significant improvements in the total training wall-clock times. Compared to SGD, Jorge improves the time to convergence by 23%, 34%, and 45% for ResNet-50, DeepLabv3, and Mask-RCNN respectively. The corresponding improvements over AdamW are even higher – 26%, 41%, and 58% (the last number is much higher since AdamW did not converge on that run). The wall-clock time improvements in these experiments highlight Jorge's applicability to small batch size training scenarios, where the overheads of a second-order optimizer cannot be masked behind network computation, making it more challenging for Jorge to beat first-order optimizers.

Table 3: Comparison of the total training time (in minutes) of Jorge with SGD and AdamW for the small batch size benchmarks (experiments run on four A100 GPUs).

| Neural Network | Batch Size | # Runs | SGD | AdamW | Jorge |
|---|---|---|---|---|---|
| ResNet-50 | 256 | 3 | $1005_{\pm 40}$ | $1052_{\pm 36}$ | $\mathbf{781}_{\pm 44}$ |
| DeepLabv3 | 64 | 5 | $217_{\pm 12}$ | $244_{\pm 0.16}$ | $\mathbf{144}_{\pm 30}$ |
| Mask-RCNN | 32 | 5 | $332_{\pm 47}$ | $438_{\pm 14}$ | $\mathbf{182}_{\pm 11}$ |

## 6 CONCLUSION AND FUTURE WORK

In this work, we introduced Jorge, an efficient, adaptive, second-order optimizer tailored to GPU platforms. We eliminated the primary computational bottleneck of computing matrix inverses in second-order optimizers by proposing a novel approximation of the preconditioner computation in Shampoo, which sidesteps the need to explicitly compute matrix inverses. Further, we proposed a single-shot hyperparameter tuning strategy, that can directly bootstrap Jorge's hyperparameters from a well-tuned SGD baseline without the need to conduct extensive tuning. We evaluated Jorge against state-of-the-art first-order optimizers – SGD and AdamW, as well as Shampoo, and we demonstrated improvements in generalization, sample efficiencies, and training wall-clock times. As future work, we plan to develop a single-shot hyperparameter bootstrapping strategy from AdamW as well. This will allow us to employ Jorge to train large language models. Additionally, we plan to develop a distributed implementation of Jorge to reduce its per-GPU memory consumption, which currently stands at 1.5–2× that of Adam (see Appendix A.7).

**Reproducibility Statement:** We are committed to enabling reproducibility of our work, as it ensures correct and transparent results. We plan to open source the code for Jorge as well as the benchmarks evaluated in this paper. Additionally, we provide a comprehensive list of all hyperparameters used in this study for each optimizer and each benchmark in Appendix A.6. The hyperparameters can be directly substituted as the arguments of SGD and AdamW shipped with PyTorch 2.0 in the "torch.optim" package. Similarly, the hyperparameters listed for Jorge will be compatible with our open source codebase.

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

# A    APPENDIX

## A.1    ENSURING VALIDITY OF THE BINOMIAL EXPANSION BY DYNAMICALLY ADJUSTING $\beta_2$

In Section 3, we mentioned that for the binomial expansion in Equation 7 to be valid, we must also ensure that $\left\| \frac{(1-\beta_2)}{\beta_2} X_L \right\| < 1$. To ensure this condition is met at every iteration, Jorge dynamically updates the EMA update parameters $\beta_2$ and $\beta_2'$ (for the right preconditioner) at each iteration. We start with the condition we want to ensure and derive a lower bound on $\beta_2$.

$$\left\| \frac{(1-\beta_2)}{\beta_2} X_L \right\| < 1 \implies \beta_2 > \frac{\|X_L\|}{\|X_L\| + 1} \tag{9}$$

Therefore, we need to set $\beta_2$ to a value higher than $\frac{\|X_L\|}{\|X_L\|+1}$ to ensure the validity of the binomial expansion. In practice, we have seen that setting $\beta_2$ equal to this quantity works well, provided we are using the Frobenius norm as our matrix norm function of choice.

Substituting the value of $\beta_2$ from Equation 9 in Equation 7 and ignoring the cubic and higher powers, gives us the complete left preconditioner update step:

$$\hat{L}_t = \left( \frac{\|X_L\| + 1}{\|X_L\|} \right)^{\frac{1}{4}} \hat{L}_{t-1} \left( I_m - \frac{1}{4} \frac{X_L}{\|X_L\|} + \frac{5}{32} \frac{X_L^2}{\|X_L\|^2} \right) \tag{10}$$

The corresponding formulation of $\beta_2'$ for the right preconditioners can be derived in a similar manner.

## A.2    COMPARING PER ITERATIONS TIMES OF JORGE, SHAMPOO, AND SGD

Here, we provide more details for the experiment mentioned in Section 3, wherein we were comparing the iteration times of Jorge with SGD and Shampoo on ResNet-50 and DeepLabv3. We use the ImageNet and MS-COCO datasets for these architectures respectively. We run ResNet-50 with a batch size of 1024 on 16 A100 GPUs. Likewise, we run DeepLabv3 with a batch size of 64 on 4 A100 GPUs. We collect the average iteration times over complete training runs and report them in Table 4. For both Jorge and Shampoo, we compute the inverse every 50 iterations. For the ResNet-50 benchmark, we observe that Jorge's iteration times are only 1% slower than SGD, whereas it is 26% faster than Shampoo! For the DeepLabv3, Jorge is only 10% slower than SGD, but a significant 21% faster than Shampoo. This experiment successfully demonstrates the computational efficiency of Jorge.

Table 4: Comparison of wallclock times per iteration (in seconds) for SGD, Jorge and Shampoo. For Jorge and Shampoo, we compute the preconditioner inverses every 50 iterations, in line with  Shi et al. (2023).

| Neural Network | Batch Size | # GPUs | SGD | Jorge | Shampoo |
|---|---|---|---|---|---|
| ResNet-50 | 1024 | 16 | 0.089 | 0.090 | 0.122 |
| DeepLabv3 | 64 | 4 | 0.33 | 0.37 | 0.47 |

## A.3    JORGE WITH GRAFTING

In Section 4, we mentioned adding grafting to Jorge, which adds a step in the weight update step of Algorithm 2. Grafting maintains the direction of the current step ($\frac{m_t}{\|m_t\|}$), but uses the magnitude of the step of a well-tuned optimizer ($\|m_{\text{SGD},t}\|$ in this case). In Algorithm 3 below, we see $m_t$ becomes $\|m_{\text{SGD},t}\| \frac{m_t}{\|m_t\|}$.

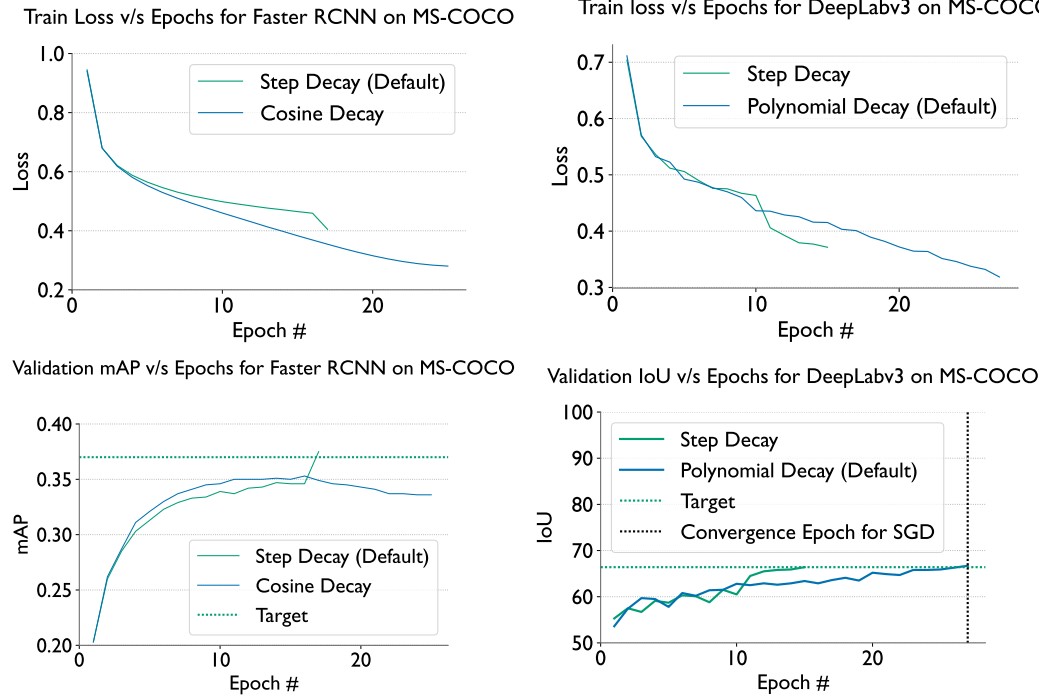

Figure 4: Comparison of different learning rate schedules with Jorge on two training tasks - Object Detection with Faster RCNN (Ren et al., 2015), and Image Segmentation with DeepLabv3 (Chen et al., 2017), both on the MS-COCO dataset (Lin et al., 2015). Both training tasks use a batch size of 64.

---

**Algorithm 3** Jorge's modified weight update rule with SGD grafting

---

1: **Update Weights:**
2: $m_t = \beta_1 m_{t-1} + (1 - \beta_1)\hat{G}_t$           ▷ Jorge weight update
3: $m_{\text{SGD},t} = \beta_1 m_{\text{SGD},t-1} + G_t$      ▷ SGD (with heavy ball momentum) weight update
4:
5: $\theta_t = \theta_t - \eta_t \|m_{\text{SGD},t}\| \frac{m_t}{\|m_t\|}$            ▷ Grafted weight update

---

### A.4 INTUITION BEHIND JORGE'S WEIGHT DECAY HEURISTIC

As mentioned in Section 4, we created a simple heuristic for setting Jorge's weight decay. Let the parameters at time step $t$ be $\theta_t$. SGD will first calculate the weight decay update as $\lambda_{\text{SGD}}\theta_t$ and then update its running estimate of the momentum using the gradients from the loss and $\lambda_{\text{SGD}}\theta_t$. Since the weight decay is a part of the running momentum estimates, the weight decay calculated at time step $t$ will influence the parameter updates at time step $t + \tau$, albeit attenuated by $\beta_{\text{SGD}}^\tau$. Therefore, the *effective* contribution of a weight decay update calculated at time step $t$ is:

$$\sum_{\tau=0}^{T-t} \beta_{\text{SGD}}^\tau \lambda_{\text{SGD}}\theta_t \approx \frac{1}{1 - \beta_{\text{SGD}}} \lambda_{\text{SGD}}\theta_t \tag{11}$$

Since we use a decoupled weight decay scheme for Jorge, the weight decay calculated at time step $t$ does not contribute to future weight updates. Therefore, to match the effective contribution of the weight decay updates in SGD, we set the weight decay penalty for Jorge to $\frac{1}{1-\beta_{\text{SGD}}} \times$ that of SGD, as shown in Equation 8.

Table 5: Hyperparameters used in this study for SGD. These are the defaults in torchvision.

| Hyperparameter | Resnet-50 (batch size 1024) | ResNet-50 (batch size 256) | DeepLab-v3 | Mask RCNN |
|---|---|---|---|---|
| Learning Rate | 0.4 | 0.1 | 0.02 | 0.02 |
| Weight Decay | 1e−4 | 1e−4 | 1e−4 | 1e−4 |
| Learning Rate Schedule | Linear warmup over 5 epochs. Then step decay at epochs 30 and 60 | Step decay at epochs 30 and 60 | Polynomial decay with 0.9 power | Step decay at epochs 16 and 22 |
| Momentum | 0.9 | 0.9 | 0.9 | 0.9 |
| Nesterov | False | False | False | False |

Table 6: Hyperparameters used in this study for Jorge.

| Hyperparameter | Resnet-50 (batch size 1024) | ResNet-50 (batch size 256) | DeepLab-v3 | Mask RCNN |
|---|---|---|---|---|
| Learning Rate | 0.4 | 0.1 | 0.02 | 0.02 |
| Weight Decay | 1e−3 | 1e−3 | 1e−3 | 1e−3 |
| Learning Rate Schedule | Linear warmup over 5 epochs. Then step decay at epochs 30 and 60 | Step decay at epochs 30 and 60 | Step decay at epochs 10 and 20 | Step decay at epochs 8 and 16 |
| Momentum | 0.9 | 0.9 | 0.9 | 0.9 |
| Preconditioner Update Freq. | 50 | 2 | 4 | 8 |

## A.5 EXPERIMENTS WITH LEARNING RATE SCHEDULES

Here we discuss the phenomenon of certain learning rate schedules leading to overfitting with Jorge, that we briefly alluded to in Section 4. Figure 4 (left) demonstrates the training loss and validation mAP curves for Faster-RCNN on MS-COCO. Notice that while the cosine schedule never reaches the target validation mAP, this is not because it sufficiently fails to minimize the training loss. In-fact, it leads to a training loss significantly lower than the step decay schedule, thereby indicating overfitting.

Similarly, Figure 4 (right) demontrates the training loss and validation IoUs for the image segmentation task with DeepLabv3. Here, the polynomial-scheduled Jorge must reach a lower training loss (loss of 0.32) than the stepwise-scheduled Jorge (0.37) to reach the same validation accuracy, once again symbolizing overfitting.

Our hypothesis for the phenomenon is that Jorge requires a high learning rate in the initial phases of training to escape sharp local minima. Due to its more accurate updates it is more prone towards falling into sharp minima compared to SGD, which might escape these because of its noisy updates. We plan to explore this phenomenon in more detail in future work.

## A.6 LIST OF HYPERPARAMETERS FOR SECTION 5

We list the hyperparameters used in this study for SGD, Jorge, and AdamW in Tables 5, 6, and 7 respectively. For Shampoo, we have used the same learning rate, weight decay and learning rate schedule as SGD, as per the recommendation of Shi et al. (2023) and enabled SGD grafting.

## A.7 ANALYSIS OF MEMORY CONSUMPTION

We mentioned in Section 6 that Jorge consumes $1.5 - 2\times$ the memory of Adam. This is because Adam uses 2 32-bit floating point optimizer states per parameter. In contrast, Jorge uses 3 (hence $1.5\times$), one each for the left preconditioner, right preconditioner, and momentum (see Algorithm 2). It becomes 4 once grafting is introduced (hence $2\times$), due to the fact that we now need to maintain

Table 7: Hyperparameters used in this study for AdamW.

| Hyperparameter | Resnet-50 (batch size 1024) | ResNet-50 (batch size 256) | DeepLab-v3 | Mask RCNN |
|---|---|---|---|---|
| Learning Rate | 0.004 | 0.01 | 0.01 | 0.01 |
| Weight Decay | 0.1 | 0.1 | 1e−3 | 1e−3 |
| Learning Rate Schedule | Cosine | Cosine | Cosine | Cosine |
| Momentum | 0.9 | 0.9 | 0.9 | 0.9 |
| $\beta$s | (0.9, 0.999) | (0.9, 0.999) | (0.9, 0.999) | (0.9, 0.999) |
| $\epsilon$ | 1e−8 | 1e−8 | 1e−8 | 1e−8 |
| amsgrad | False | False | False | False |

the momentum for SGD as well. This is a major limitation of our method, and one which we plan to fix with a distributed implementation.

