# OpenReview forum: "Jorge: Approximate Preconditioning for GPU-Efficient Second-Order Optimization"
_ICLR.cc/2024/Conference — Submitted to ICLR 2024_

### Official Review · Reviewer_yz1B · 2023-10-22

**Soundness:** 2 fair
**Presentation:** 4 excellent
**Contribution:** 3 good
**Rating:** 5
**Confidence:** 5

**Summary:**

The paper presents a method to approximate the preconditioners used in the Shampoo optimizer by using a Taylor series. The authors show that this method has performance comparable to that of Shampoo in terms of number of steps, but since they do not need to compute inverses and fourth roots, the algorithm can run much faster than Shampoo. They also show that the performance gains obtained from second order optimization continue to hold with the approximation.

**Strengths:**

The main result of the paper is in Section 3, where the authors describe their approximation. This is very clearly described --- although for some reason the next step --- using X_L / || X_L || instead of (1 - b)/b X_L to ensure convergence --- is relegated to the appendix.
The theory and experimental results are well described.

**Weaknesses:**

The idea is quite straightforward (but maybe that is not a weakness).

The equality on line -2 on page 4 is not valid:  (A + B)^r =/= A^r (I + A^{-1} B)^r, unless A and B commute. Is there a way to fix the equation?

**Questions:**

The authors main claim is that the Taylor approximation is sufficient to compute the preconditioners --- do you have some direct evidence of this (i.e. some illustration of the difference between the original preconditioners and the Jorge approximations)?

In your paper please do include the Shampoo metrics as a column in all your tables, since you already have them.

In most Shampoo implementations, the matrix roots are also computed via an iterative process that only involves matrix multiplications. How does your approximation compare with this process, for example by limiting the number of iterations to a fixed value?

You say that (1 - b)/b X_L is chosen adaptively to ensure convergence of the Taylor approximation --- what values of b do you get as a result? Note that if b is small than effectively the previous gradients will be forgotten very quickly, which may lead to instability (so the b also serves a different purpose than the one you use it for).

---

> ### Author Response · Authors · 2023-11-23
> **Response to Reviewer yz1B**
>
> Thank you for your valuable comments and constructive feedback, Reviewer yz1B.
>
> > "The authors main claim is that the Taylor approximation is sufficient to compute the preconditioners. Do you have some direct evidence of this (i.e. some illustration of the difference between the original preconditioners and the Jorge approximations)? "
>
> While Jorge computes an approximation of Shampoo's preconditioner calculation, our experimental results, as illustrated in Figure 3 (left), demonstrate that it matches the sample efficiency of Shampoo. This provides empirical evidence supporting the sufficiency of our approximation in computing the preconditioners. We acknowledge the suggestion for a more fine-grained analysis, such as a direct comparison between the approximate and exact preconditioners, and we plan to incorporate this in our final manuscript to further strengthen this claim.
>
> > "In most Shampoo implementations, the matrix roots are also computed via an iterative process that only involves matrix multiplications. How does your approximation compare with this process, for example by limiting the number of iterations to a fixed value?"
>
> The implementation of Shampoo we compared with [1] includes support for an iterative scheme known as the Coupled Inverse Newton's iteration. We acknowledge the importance of comparing Jorge's iteration times with this scheme, and we plan to incorporate this comparison in our final manuscript for a more comprehensive evaluation.
>
> > "You say that (1 - b)/b X_L is chosen adaptively to ensure convergence of the Taylor approximation .. "
>
> While a very low value of b can lead to gradients being forgotten quickly, we did not observe any convergence issues in any of our experiments (Figures 2 and 3). We agree that it might be useful to have a hyperparameter that serves as a lower bound on the value that our adaptive scheme can assign to b.
>
> > The equality on line -2 on page 4 is not valid: (A + B)^r =/= A^r (I + A^{-1} B)^r, unless A and B commute. Is there a way to fix the equation?
>
> Thank you for bringing this to our attention. You are correct, and we appreciate your observation. The equality (A + B)^r =/= A^r (I + A^{-1} B)^r is valid only when A and B commute. We acknowledge this limitation and will make the necessary clarification.
>
>
> [1] Hao-Jun Michael Shi, Tsung-Hsien Lee, Shintaro Iwasaki, Jose Gallego-Posada, Zhijing Li, Kaushik Rangadurai, Dheevatsa Mudigere, and Michael Rabbat. A distributed data-parallel pytorch implementation of the distributed shampoo optimizer for training neural networks at-scale, 2023.
> .

---

### Official Review · Reviewer_LZ4W · 2023-10-27

**Soundness:** 2 fair
**Presentation:** 3 good
**Contribution:** 2 fair
**Rating:** 5
**Confidence:** 4

**Summary:**

This paper proposes a new second-order optimizer for deep learning.
The proposed method, Jorge, based on Shampoo, but does not require matrix inversion when applying gradient conditioning.
The key contribution in Jorge is to approximate matrix inversion in Shampoo with binomial series expansion, and keep the first and second item.
Owing to the approximation, Jorge can be done efficiently on GPUs in terms of wall-clock time.
Experiments show that Jorge achieves faster wall-clock training compared to SGD and Adam, as well as Shampoo.

**Strengths:**

This paper targets a practical and important problem of reducing the complexity of second-order optimizers for deep learning, especially for large models. Several contributions are highlighted below

**1**, The presentation is clear.

The way the authors present Jorge is very easy to understand. By having a point-to-point comparison with Shampoo (Alg 1 and 2), I can quickly understand how Jorge adapts Shampoo and how matrix inversion is avoided when applying pre-conditioning.

**2**, Jorge reduces efforts for optimizing training hyperparameters.

Finding training hyperparameters for second-order methods is a practical and important problem. The way Jorge learns hyperparameters (learning rate, weight decay, scheduler) is important to reduce overall training time, especially for large models.

**3**, The evaluations cover different tasks.

The authors demonstrate the efficacy of Jorge on different tasks, which shows Jorge not only achieves faster convergence, but also gives comparable validation accuracy as SGD.

**Weaknesses:**

**1**, Lack of analysis for the proposed method

While the authors propose a practical solution that avoids computing matrix inversion, it lacks the necessary analysis:

**1)**, The authors need to analyze computation and memory complexity for Jorge.

While Jorge avoids to compute matrix inversion, it also introduces additional operations (lines 5-8 in Algorithm 2). It is necessary to analyze the complexity of these additional matrix multiplications compared to matrix inversion in line 11 in Algorithm 1.
I noticed the authors provide a wall-clock comparison between Jorge and Shampoo. However, it would be more insightful to have a theoretical complexity analysis.

**2)**, It is unclear how accurate the approximation is.

In Eq(7), high-order items are removed. It is unclear how much error is introduced after the approximation. And importantly, how is the error propagated with Eq(4)?
While it is impractical to analyze the approximation for highly non-convex NNs, it is possible to analyze in a simplified case, like simple convex functions.
Therefore, I would suggest the authors do an analysis on the effectiveness of the approximation.

**3)**, No convergence analysis.

Besides empirical analysis, it is also important to analyze convergence and show how the approximation and related parameters affect the convergence speed.

**2**, Comparison between Jorge and baseline methods needs further justification.

In experiments, the authors use step learning rate decay for Jorge, SGD, and Shampoo, but use a cosine decay for Adam. It is difficult to compare convergence speed with different learning rate schedulers. I understand different optimizers might prefer different schedulers. However, it would be more straightforward if compare optimizers with the same scheduler.

**1)**, According to Table 2, AdamW achieves validation accuracy close to Jorge. However, there is no convergence plot like Figure 2 showing the convergence speed of Adam compared to Jorge. It would be great if the authors also provide such a plot in the paper.

**3**, Lack of necessary ablation studies.

It would be good if the authors provide more ablation studies.
For instance, how does the preconditioner update frequency affect convergence and final model performance?

**4**, Need more baselines.

For reducing the complexities of using second-order methods, optimizers such as K-FAC [1] and mL-BFGS [2] also target the same goal. It would be good if the authors compare Jorge against these methods in terms of per-iteration and wall-clock convergence performance.

[1] George, T., Fast approximate natural gradient descent in a kronecker factored eigenbasis. NeurIPS'18

[2] Niu, Y., mL-BFGS: A Momentum-based L-BFGS for Distributed Large-scale Neural Network Optimization. TMLR'23.

**Questions:**

**1**, According to Algorithm 2, Jorge introduces additional matrix multiplications compared to SGD. However, in Figure 2, it seems per-iter running time of Jorge is almost the same as SGD. Can the authors explain that?

---

> ### Author Response · Authors · 2023-11-23
> **Response to Reviewer LZ4W**
>
> Thank you for your valuable comments and constructive feedback, Reviewer LZ4W.
>
> > Lack of analysis for the proposed method
>
> We value the reviewer's suggestion and will address it by incorporating a theoretical complexity analysis for computation and memory in our final manuscript.
>
> > Convergence Analysis
>
> To the best of our knowledge, convergence analysis for adaptive optimizers like Shampoo has primarily been limited to convex functions. While we can conduct convergence analysis for Jorge in the context of convex functions, its utility for neural network training—characterized by highly non-convex loss landscapes—remains uncertain.
>
> > "In experiments, the authors use step learning rate decay for Jorge, SGD, and Shampoo, but use a cosine decay for Adam. "
>
> We used the cosine decay schedule with AdamW on the findings of its authors, who demonstrated its superior generalization compared to step decay and constant learning rate schedules [1].
>
> >  "there is no convergence plot like Figure 2 showing the convergence speed of Adam compared to Jorge."
>
> In Figure 3, we present a comparison of the convergence speed between AdamW and Jorge in terms of the number of epochs. The corresponding wall clock times for convergence in these experiments are detailed in Table 3. Hence, we have assessed the convergence speeds of Jorge and AdamW from both the perspective of epochs and wall clock times.
>
> We plan to augment our final manuscript by incorporating additional comparisons with more second-order methods and including ablation studies to provide a more comprehensive analysis.
>
>
> [1]  Decoupled Weight Decay Regularization, Ilya Loshchilov, Frank Hutter

---

### Official Review · Reviewer_9qvx · 2023-10-31

**Soundness:** 1 poor
**Presentation:** 4 excellent
**Contribution:** 2 fair
**Rating:** 3
**Confidence:** 2

**Summary:**

Building off of Shampoo, the authors introduce a new second-order SGD optimizer, called Jorge.  In order to speed up the original second-order SGD optimizer, the author's avoid all matrix-inverse calculations used in Shampoo.  Jorge approximates all such matrix inverses, leading to a second-order optimizer whose preconditioner only relies on GPU-friendly operations (i.e., matmul and matrix additions).  The authors then show how Jorge's hyperparameters are bootstrapped, via grafting given a well-tuned, prerun SGD optimizer.  Jorge is then evaluated over 3 vision CNN architectures (ResNet-50, Mask-RCNN, and DeepLabv3) and datasets (ImageNet and MS-COCO 2017), and compared to 2 first-order solvers (AdamW and vanilla SGD) and 2 implementations (serial and distributed) of the second-order Shampoo solver.

**Strengths:**

The work tackles an important problem; first-order solvers remain the go-to optimizers for DNN architectures across application domains.  A truly efficient second-order solver has the potential to decrease the significant levels of overhead currently being incurred by training LLMs and other massive-scale, data hungry models (e.g., diffusion models).

The authors did an excellent job in their writing, the paper was easy to follow.  Furthermore, the description of Jorge and its juxtaposition to the original Shampoo algorithm were well done.

**Weaknesses:**

While the derivation of Jorge is interesting and the focus on GPU-friendly computations makes sense, the paper contains several causes for concerns.  I look forward to hearing the authors' replies to the following issues.

# Comparison to other second-order solvers
Evaluation to other related methods is limited, where only a single second-order optimizer is compared to.  The authors reference the recent SGD L-BFGS papers, but it should also be explained why these methods aren't compared to (i.e., implementations of these SGD L-BFGS have not been released).  However, the following second-order SGD optimizer should be compared to in the paper:
- Yao, Zhewei, et al. "Adahessian: An adaptive second order optimizer for machine learning." proceedings of the AAAI conference on artificial intelligence. Vol. 35. No. 12. 2021.

# Reported metrics, performance of/fair-comparison to AdamW
> We borrow AdamW hyperparameters for the imagenet benchmarks from Heo et al. (2021).  The complete list of all hyperparameters used in this study can be found in Appendix A.6.

For ResNet50+Imagenet in the Heo et al. paper, both AdamW and SGD achieve ~76% accuracy, whereas in the current paper, AdamW saturates at ~70% (in Figure 2).  Furthermore, Figure 2 clearly shows SGD and Jorge benefiting from the authors decay schedule, which is not a fair comparison for AdamW (i.e., SGD is further well-tuned, while AdamW is not).  A fairer comparison for all methods would to use the same hyperparameters for both SGD and AdamW from Heo	et al. (2021), while also determining the cause of performance loss for	AdamW observed in this work.

Furthermore, Jorge requires running SGD prior to each step (due to Jorge's use of grafting SGD).  However, Jorge reported runtime in Table	3 is lower than	SGD itself.  Thus, Jorge's reported timings seem inaccurate; because Jorge is dependent	on grafting SGD, its runtime should reflect this.

# Lack of Adam support and transformer architecture evaluations
Given the ubiquity and importance of transformer architectures used across applications (not only in NLP, but vision as well), Jorge's impact on transformer training should also be demonstrated.  A smaller issue is, similarly, Adam/AdamW is widely used, even for vision (e.g., vision transformers, ViT), and the inclusion of this optimizer for bootstrapping Jorge's hyperparameters is warranted for current deep learning.

**Questions:**

- For Table 3, please include the (average) number of epochs required to achieve target performance.

- > Interestingly, simply switching to the step decay schedule with 2 decay steps (reducing the learning rate by 10× at each step) at one-third and two-thirds of the total training epochs (total epochs same as that of the tuned SGD baseline) resolves this issue

Note that similar behavior and decay schedule was used in Zhang et al "Lookahead Optimizer: k steps forward, 1 step back".

- > For example, SGD remains the optimizer of choice in computer vision due to its better final validation accuracy, even though Adam converges faster initially.

This is a very debatable claim, please revise.  In particular, Adam is widely used as the out-the-gate optimizer for vision transformers (e.g., ViT) as well as CLIP.  Furthermore, even in extensive benchmark studies, mixed results refute this statement, e.g., see:
Kumar, Ananya, et al. "How to fine-tune vision models with sgd." arXiv preprint arXiv:2211.09359 (2022).

---

> ### Author Response · Authors · 2023-11-23
> **Response to Reviewer 9qvx**
>
> Thank you for your valuable comments and constructive feedback, Reviewer 9qvx.
>
> > Comparison to other second-order solvers
>
> We acknowledge the reviewer's observation regarding the limited comparison with other second-order optimizers in our current manuscript. We agree that a more comprehensive evaluation involving a broader set of second-order optimizers is necessary. We plan to address this gap in our final draft by including comparisons with additional second-order optimizers like Adahessian and KAISA.
>
> > Concerns about AdamW
>
> We will investigate why AdamW did not reach the target validation accuracy of 76% for ResNet50 with a batch size of 1024. Interestingly, we were able to reach 76% validation accuracy with AdamW with a batch size of 256 (Figure 3, left). We opted for the cosine learning rate schedule with AdamW based on the findings of its authors, who demonstrated its superior generalization compared to step decay and constant learning rate schedules [1].
>
> > Furthermore, Jorge requires running SGD prior to each step (due to Jorge's use of grafting SGD). However, Jorge reported runtime in Table 3 is lower than SGD itself.
>
> Indeed, Jorge's per-iteration times may be slower than SGD due to the grafting process. However, Table 3 does not display per-iteration times; instead, it illustrates the total wall clock time for convergence. Notably, Jorge necessitates 25-40% fewer iterations than SGD to converge to a target validation metric, as illustrated in Figure 3. Consequently, the overall wall clock times are substantially smaller for Jorge, despite being slower on a per-iteration basis
>
> > Lack of Adam support and transformer architecture evaluations
>
> We recognize the significance of showcasing Jorge's performance on transformer architectures, given their widespread application. To enhance the experimental section of our paper, we plan to include results on transformer architectures in the final manuscript. Moreover, considering the prevalent use of AdamW, particularly in transformer training, we intend to implement a bootstrapping strategy to derive hyperparameters from AdamW in our experimental evaluations.
>
> We will include the epochs for convergence in Table 3, which can be inferred from Figure 3 as well.
>
> We will adjust the claim regarding SGD being better than AdamW to be more nuanced and considerate.
>
>
> [1] Decoupled Weight Decay Regularization, Ilya Loshchilov, Frank Hutter

---

### Official Review · Reviewer_k7BV · 2023-11-09

**Soundness:** 2 fair
**Presentation:** 3 good
**Contribution:** 2 fair
**Rating:** 5
**Confidence:** 4

**Summary:**

The paper proposes a compute-efficient approximate second-order optimization algorithm named Jorge for training deep neural networks. Jorge has a similar per-iteration speed with SGD but has a faster convergence speed in terms of the # of iterations than SGD. Jorge is also hyper-parameter friendly as it can enjoy the well-tuned hyperparameters from SGD. Some experiments show that Jorge is faster than SGD, AdamW, and Shampoo on some CV tasks.

**Strengths:**

- The proposed approximate second-order optimizer seems novel.
- Comprehensive experiments to show the effectiveness of Jorge.
- The paper is well-written.

**Weaknesses:**

- Some important baselines are missing, e.g., KAISA [ref1], Eva [ref2], and MKOR [ref3].
- Jorge is built based on the key idea of Shampoo, but Jorge seems more hyper-parameter friendly than Shampoo as claimed in the paper. It is unclear why Jorge has such a property.

[ref1] Kaisa: an adaptive second-order optimizer framework for deep neural networks, SC'21.
[ref2] Eva: practical second-order optimization with Kronecker-vectorized approximation, ICLR'23.
[ref3] MKOR: Momentum-Enabled Kronecker-Factor-Based Optimizer Using Rank-1 Updates, NeurIPS'23.

**Questions:**

- Could you include some recent compute-efficient second-order algorithms like KAISA (with a large update frequency), Eva, and MKOR for comparison?
- What is the update frequency used in Shampoo when comparing wall-clock time? Is possible to use a second-order update interval for Shampoo such that it runs at a similar speed to Jorge to achieve the target accuracy?
- Could you explain why Jorge can enjoy well-tuned hyper-parameters from SGD while Shampoo cannot?
- Is Jorge possible to apply to NLP tasks like LLM pertaining or fine-tuning?

---

> ### Author Response · Authors · 2023-11-23
> **Response to Reviewer k7BV**
>
> Thank you for your valuable comments and constructive feedback, Reviewer k7BV.
>
> > Could you include some recent compute-efficient second-order algorithms like KAISA (with a large update frequency), Eva, and MKOR for comparison?"
>
> We agree that the experimental evaluation of our paper can be strengthened by including comparisons with second order optimizers other than Shampoo. We plan to include these in the final draft of our manuscript.
>
> > What is the update frequency used in Shampoo when comparing wall-clock time?
>
> In line with Shi et al. [1], we use a second-order update interval of 50 for Shampoo.
>
> > Is possible to use a second-order update interval for Shampoo such that it runs at a similar speed to Jorge to achieve the target accuracy?
>
>  It is possible to adjust the second-order update interval for Shampoo to achieve a similar iteration time as Jorge. For example, for ResNet-50 on 16 GPUs (as shown in Table 4, row 1), setting Shampoo's update frequency to approximately 1500 would result in similar iteration times as Jorge with an update frequency of 50. However, it's crucial to note that such large update intervals for Shampoo may impact convergence significantly, as demonstrated by Anil et al. [2] (Figure 4c of their paper).
>
> > Could you explain why Jorge can enjoy well-tuned hyper-parameters from SGD while Shampoo cannot?
>
> We would like to clarify that Shampoo, as demonstrated by Shi et al. [1] and Anil et al. [2], can indeed borrow certain hyperparameters from well-tuned first-order optimizers such as SGD and Adagrad. Our contribution lies in the specific insights we provide in the paper regarding the determination of optimal weight decay and learning schedules from SGD. As highlighted in our study, this process is nuanced and not straightforward, and our findings contribute to the understanding of how these hyperparameters can be effectively set for Jorge.
>
> > Is Jorge possible to apply to NLP tasks like LLM pertaining or fine-tuning?
>
> We are currently actively investigating the applicability of Jorge to NLP tasks, including language model pretraining and fine-tuning. As part of our ongoing research, we aim to provide a comprehensive evaluation of Jorge's performance in the context of NLP applications.
>
> [1] Hao-Jun Michael Shi, Tsung-Hsien Lee, Shintaro Iwasaki, Jose Gallego-Posada, Zhijing Li,
> Kaushik Rangadurai, Dheevatsa Mudigere, and Michael Rabbat. A distributed data-parallel pytorch implementation of the distributed shampoo optimizer for training neural networks at-scale, 2023.
>
> [2] Rohan Anil, Vineet Gupta, Tomer Koren, Kevin Regan, and Yoram Singer. Scalable second order
> optimization for deep learning, 2021.

---

### Meta-Review · Area_Chair_6CUd · 2023-12-09

**Metareview:**

This paper proposes a new second-order optimizer, called Jorge, which is heavily influenced by the prior work, Shampoo. The advantages of Jorge is that it does not require matrix inversions in Shampoo when applying gradient conditioning. This is done using approximations from the binomial series expansion.

On one hand, although some justification are provided, the paper lacks any theoretical analysis. On the other, the empirical study is very limited and lacks several key aspects expected from a mainly empirical paper, e.g., comparison related approaches, adequate ablation studies, problem sets, etc.

**Justification For Why Not Higher Score:**

The paper lacks any theoretical analysis and the empirical study is very limited.

**Justification For Why Not Lower Score:**

N/A

---

### Decision · Program_Chairs · 2024-01-16

Reject